# The SED-GIH: A Single-Item Question for Assessment of Stationary Behavior—A Study of Concurrent and Convergent Validity

**DOI:** 10.3390/ijerph16234766

**Published:** 2019-11-28

**Authors:** Lena V. Kallings, Sven J. G. Olsson, Örjan Ekblom, Elin Ekblom-Bak, Mats Börjesson

**Affiliations:** 1Åstrand Laboratory of Work Physiology, The Swedish School of Sport and Health Sciences, GIH, 114 86 Stockholm, Sweden; rottninge@yahoo.se (S.J.G.O.); orjan.ekblom@gih.se (Ö.E.); elin.ekblombak@gih.se (E.E.-B.); 2Family Medicine, Department of Public Health and Caring Sciences, Uppsala University, 751 22 Uppsala, Sweden; 3Institute of Neuroscience and Physiology and Institute of Food, Nutrition and Sport Science, Göteborg University, 405 30 Gothenburg, Sweden; mats.brjesson@telia.com; 4Sahlgrenska University Hospital/Östra, 416 50 Gothenburg, Sweden

**Keywords:** sedentary behavior, accelerometry, surveys and questionnaires, validation studies, public health, measurement

## Abstract

The unfavorable health consequences of prolonged time spent sedentary (stationary) make accurate assessment in the general population important. However, for many existing questionnaires, validity for identifying stationary time has not been shown or has shown low validity. This study aimed to assess the concurrent and convergent validity of the GIH stationary single-item question (SED-GIH). Data were obtained in 2013 and 2014 from two Swedish cohorts. A total of 711 men and women provided valid accelerometer data (Actigraph GT3X+) and were included for concurrent validity analyses. A total of 560 individuals answered three additional commonly used sedentary questions, and were included for convergent validity analysis. The SED-GIH displayed a significant correlation with total stationary time (r_s_ = 0.48) and time in prolonged stationary time (r_s_ = 0.44). The ROC analysis showed an AUC of 0.72 for identifying individuals with stationary time over 600 min/day. The SED-GIH correlated significantly with other previously used questions (r = 0.72–0.89). The SED-GIH single-item question showed a relatively high agreement with device-assessed stationary behavior and was able to identify individuals with high levels of stationary time. Thus, the SED-GIH may be used to assess total and prolonged stationary time. This has important implications, as simple assessment tools of this behavior are needed in public health practice and research.

## 1. Introduction

Sedentary behavior (SED), even after adjustment for physical activity (PA) level, has been associated with increased risk for cardiovascular disease, metabolic syndrome, type 2 diabetes, and some forms of cancer [1,2], as well as total and cardiovascular mortality [2,3,4]. Reducing total time spent being sedentary, as well as breaking up prolonged sedentary periods, is important to counteract some of the negative effects of the cardiometabolic risks of sedentary behavior [5,6,7,8].

With the development of new assessment tools such as accelerometry, for approximating physical activity patterns, the definitions of sedentary behavior have gradually changed. Most existing questionnaires used for PA assessment identify individuals’ self-reported physical inactivity. The difference between sedentary behavior (SED) and physical inactivity has been defined for accelerometer data, with SED being activity below a certain level of accelerometer counts at the low end of the inactivity spectrum. However, the definition of SED also includes posture, i.e., sitting, reclining, or lying position [9], and accelerometers are designed to record ambulation and not to distinguish posture. Recently, the SBRN (Sedentary Behavior Research Network) proposed the term “stationary behavior” for any waking behavior done while lying, reclining, sitting, or standing with no ambulation, regardless of the energy expenditure [9]. Separating stationary from ambulatory behaviors seems to be justified also from a metabolic perspective, since walking, but not standing, has been reported to alleviate consequences from prolonged sitting [10,11]. Moreover, time spent in sedentary behaviors seem to change differently over time in different domains [12].

Many existing studies of the association between sedentary behavior and health have relied on non-validated questions or modified versions of questions. There is a need for a feasible assessment tool to identify stationary behavior. The validity for questions aimed at assessing stationary time has not been published.

Therefore, we developed a global question, with the aim of identifying people with high (>10 h per day) levels of stationary behavior. Results from previous studies were used in the design of the question [13,14]. The rationale for choosing categorical answer alternatives was that such answer modes have been shown to provide superior validity compared to open answers when asking for level of PA [13].

The main aim of this study was to assess the concurrent validity for stationary behavior, using accelerometry as criterion measurement, of the new SED-GIH single-item question. The secondary aim was to assess the convergent validity of the SED-GIH and other commonly used and validated questions for assessment of sedentary behavior.

## 2. Materials and Methods

### 2.1. Study Sample and Design

This study was based on data from 711 adults from two cross-sectional samples that we collected in 2013–2015 [13,15,16]. One was a random sample from the Swedish population aged 20–65 years (*n* = 465), and the other was made up of employees of a large Swedish company with work sites across Sweden (*n* = 246). Both samples answered the self-administrated LIV 2013 questionnaire, which contains the SED-GIH question and other questions assessing sedentary behavior (see below). Inclusion criteria for this study were having answered the SED-GIH and providing valid accelerometer data. Of these, 560 answered three additional commonly used sedentary questions (see below), which were included for convergent validity analysis of the SED-GIH question.

All subjects gave their informed consent for inclusion before they participated in the study. The study was conducted in accordance with the Declaration of Helsinki, and the protocol was approved by the Regional Ethical Review Board in Stockholm, Sweden (2012/1338-31).

### 2.2. Self-Reported Stationary Behavior

The background for the formation of the present SED-GIH question was other existing questionnaires and data from previous studies, indicating higher validity for categorical answers alternatives compared to open answer options when assessing PA level [13]. The SED-GIH question reads “How much time do you sit a normal day, excluding sleep?” with seven categories presented to the respondent, ranging from “Virtually all day”, “13–15 h”, “10–12 h”, “7–9 h”, “4–6 h”, “1–3 h”, to “Never”. Thus, this question uses total sitting time as an indirect means (proxy) for all stationary time. Although these entities are not equal, the theory was that sitting time is easier to recall correctly, compared to all stationary time.

To evaluate the convergent validity, we used three other commonly used questions on sedentary behavior. The first was a question asking about the amount of time spent sitting during the course of most days of the week, with five categories ranging from sitting almost all of the time to almost none of the time (Katzmarzyk) [3]. Secondly, the sitting item from the short form of the International Physical Activity Questionnaire (IPAQ) was used [17]. The third question was a domain-specific sitting question that asked for sitting each day on weekdays and weekend days across five domains (transportation, at work, watching TV, using computer at home, and during leisure time. TV and computer time excluded) [18,19]. Only the sum of all of the weekday modes was used in the present study (Marshall).

### 2.3. Accelerometer

Actigraph GT3X+ accelerometers (ActiGraph LLC, Pensacola, FL, USA) were used to assess stationary behavior. These devices have been shown to have a limited validity for separating sitting from standing [20], but have been proven valid for measuring the duration, frequency, and intensity of PA and stationary behavior [21,22]. The participants were asked to wear the accelerometer on the right hip during waking hours for seven consecutive days, but not when swimming, bathing, etc.

The accelerometers and data files were handled with the software ActiLife 6.11 (ActiGraph LLC, Pensacola, FL, USA). Accelerometer data were analyzed as uniaxial data (vertical axis). The low frequency extension filter, 30 Hz sample rate, and idle sleep mode was utilized. Raw data were down-sampled to a 60 s epoch, and PA data were expressed in counts per minute (cpm). Non-wear time was defined as ≥60 min of consecutive zero cpm with an allowance of two minutes of ≤100 cpm [23]. After exclusion of non-wear time, ≥4 days with ≥10 h of data per day were required for inclusion in analysis [23,24].

Time in stationary behavior (STA) was defined as 0–99 cpm and moderate and vigorous physical activity (MVPA) as cpm > 2020 [25]. The mean total STA time per day was calculated as the sum of STA on all valid days divided by the number of valid days. Prolonged STA time was defined as periods of STA in bouts of ≥20 consecutive minutes below 100 cpm, with no allowance for interruption above threshold (STA bouts) [8,26,27]. Average prolonged stationary time per day (prolonged STA) was obtained by dividing the sum of all STA bouts on all valid days by the number of valid days.

### 2.4. Statistical Analysis

All statistical analyses were performed in IBM SPSS (IBM Corp. Released 2016. IBM SPSS Statistics for Windows, Version 24.0. Armonk, NY, USA: IBM Corp.).

When analyzing SED-GIH and the question developed by Katzmarzyk, the answer alternatives were re-arranged in order from low to high sitting time. Few participants answered the highest answer categories, and to get enough power, the two alternatives were merged respectively. In SED-GIH, the two categories with lowest sitting time were also merged due to few participants in those categories.

Studies have shown that non wear time is important to consider, especially for sedentary behavior, since non wear time in the morning and evening hours affects sedentary time [28]. Different strategies have been described to statistically adjust for wear time [29,30,31]. In this study, the standardized wear time was set to 16 h (allowing for an 8 h sleep period per full day).

Device-assessed time spent stationary was compared using a Mann–Whitney U-test across strata of gender, age (dichotomized at median = 50.0 years), standardized time spent in MVPA (dichotomized at median 44.0 min/day), SES (i.e., education dichotomized for university degree vs. less), and waist circumference (dichotomized at median 88 cm for women and 102 for men).

Each participant was assigned to one of three categories according to the agreement of the reported sitting time and the assessed time spent stationary. Although being different entities, categories were named “under-reporting”, “correct reporting”, and “over-reporting”. For the 0–3 h, 4–6 h, 7–9 h, and 10+ h response categories, correct reporting was assumed if the device-assessed stationary time fell between 0–209 min, 210–389 min, 390–569 min, and 570 min+, respectively.

Descriptive data are presented as both observed and standardized time. In all analyses, standardized stationary time was used. Concurrent validity was assessed using Spearman’s rho, comparing self-reported sedentariness in categories and minutes per week with accelerometer data (total stationary time and prolonged stationary time). The associations were interpreted as weak (Spearman’s rho < 0.1), modest (Spearman’s rho 0.1–0.3), strong (Spearman’s rho 0.5–0.8), or very strong (Spearman’s rho 0.8–1.0) [32]. Receiver operating characteristic (ROC) analyses were used to assess the ability of the SED-GIH to correctly classify participants as having high levels (over 10 h per day) of STA behaviors. The cut-off value was chosen based on relationships between accelerometer-derived sedentary time and all-cause mortality or cardiometabolic risk factors [33,34,35,36]. ROC results are presented as area under the curve (AUC) with 95% confidence intervals and sensitivity and specificity. Spearman’s rho and gamma statistics were used to analyze convergent validity between GIH-SED and the other questions regarding sedentary behavior.

## 3. Results

### 3.1. Sample Characteristics

The concurrent validity sample consisted of 711 individuals, 69% women, with a mean age of 48 ± 12 years. Men were older than women, averaging 50 ± 12 years vs. 48 ± 12 years (*p* = 0.018). Mean BMI was 25.7 ± 4 kg/m^2^, with a wide span (min–max 17.1–42.0). The convergent validity sample consisted of 560 individuals, 66% women, with a mean age of 48 ± 12 years, with men being older than women, 50 ± 12 years vs. 47 ± 12 years (*p* = 0.006). Men reported more stationary behavior according to both the SED-GIH question and the three additional questions compared to women, and also spent more time stationary (both total time and in prolonged bouts >20 min) according to the accelerometer assessment (Table 1).

### 3.2. Concurrent Validity of SED-GIH Using Accelerometer as Reference

There was a moderate relationship between stationary time according to the SED-GIH and standardized total stationary time assessed by accelerometers, as shown in Figure 1. Considerable overlap in assessed time was observed across answer categories. SED-GIH correlated significantly (*p* < 0.001) with total standardized stationary time (r = 0.48) for the total sample, as well as among women (r_s_ = 0.50) and men (r_s_ = 0.36). There was a statistically significant (*p* < 0.001) difference in standardized total stationary time across the different answer categories of SED-GIH between all categories except between 7–9 h and 13 h and 10–12 h and 13 h.

SED-GIH correlated significantly (*p* < 0.001) with standardized prolonged stationary time assessed by accelerometer (r = 0.44) in the whole sample (Figure 2), and in women (r = 0.48) and men (r = 0.28) analyzed separately. There was a statistically significant (*p* < 0.001) difference in standardized, prolonged stationary time across the different answer categories of SED-GIH, with differences between category 0–3 h and all other categories, and between category 4–6 h and all other categories.

Within each response category of the SED-GIH (0–3, 4–6, 7–9, and 10–13+ h), device-assessed stationary time was compared across strata of sex, MVPA, SES, age, and waist circumference (Figure 3). Stationary time was significantly lower among participants with standardized MVPA time above median in three of four response categories. In the response category 4–6 h, male participants had a significantly higher time spent being stationary, compared to females.

As shown in Table 2, assessed mean total stationary time was considerably higher than that reported in the lower response categories of SED-GIH, implying a systematic under-reporting. In higher answer categories, reported time was lower than assessed, resulting in a partial over-reporting. When analyzed in relation to different strata, sex, MVPA, and age affected the frequencies of under-reporting, correct reporting, and over-reporting (Table 2).

ROC analysis for SED-GIH, using 10 h of standardized total stationary time as indicator of a hazardous stationary time, revealed an AUC of 0.72 (95% CI: 0.68–0.76). Sensitivity analyses showed only marginal differences if using 8 h (AUC 0.73, 95% CI: 0.69–0.77) or 6 h (AUC 0.73, 95% CI: 0.62–0.83). A cut-off value for the SED-GIH identified answer Alternative 7–9 h as the point estimate that generated the strongest combination of sensitivity (67%) and specificity (68%) for stationary behavior.

### 3.3. Convergent Analysis

In the convergent validity sample, SED-GIH and the other three sedentary questions correlated significantly (*p* < 0.001) with both standardized total stationary time and standardized prolonged stationary time assessed by accelerometer (Table 3).

Moreover, SED-GIH correlated highly with the other three questions regarding sedentary behaviors. (Table 3). ROC analyses for the identification of individuals with more than 10 h/day of accelerometer-based-STA time revealed that SED-GIH (AUC 0.71, 95% CI: 0.66–0.75) had similar diagnostic ability to the commonly used questions by Katzmarzyk (AUC 0.73, 95% CI: 0.68–0.77), IPAQ (AUC 0.70, 95% CI: 0.66–0.75), and Marshall (AUC 0.72, 95% CI: 0.67–0.77).

## 4. Discussion

The main finding from this study was that a new single-item global question of total sitting time can be used as an indirect mean or proxy for total stationary time. The SED-GIH had a moderate concurrent validity with total stationary time and prolonged bouts of stationary time, as assessed by accelerometer. In addition, the SED-GIH classified individuals to a fair extent as having or not having high levels of stationary behaviors.

In addition, the SED-GIH correlated significantly (*p* < 0.001) with three other commonly used questions (r_s_ between 0.72–0.89), showing its convergent validity. The other questionnaires also correlated significantly (*p* < 0.001) with total stationary time and prolonged stationary time assessed by accelerometer.

Time spent in MVPA, gender, and age were all significant correlates of over- or under-reporting. This showed that for two groups of individuals giving the same responses, actual time spent stationary could be assumed to differ significantly between strata in these variables.

This study was, to our knowledge, unique, as the concurrent validation of questions regarding sitting were correlated with both prolonged stationary time (≥20 min bouts) and total stationary time assessed by accelerometer. That SED-GIH also correlates with prolonged stationary time increases its usefulness, as prolonged stationary time is increasingly important to assess in clinical practice.

Our study fulfilled the call to conduct studies with several questionnaires for sedentary behavior evaluated at the same time in the same population [37]. Recently, one validation study of several PA questions, including one open question of total sitting time, also used prolonged sitting. The concurrent validity was low to moderate, with a correlation for the sitting item and accelerometer-assessed total sedentary time of (r_s_ = 0.3) and prolonged sedentary time (r_s_ = 0.2) [36], showing lower validity than the SED-GIH question of this study. Another clinical assessment tool for sedentary behavior has been validated against accelerometry in a primary health care population and the correlation with total sedentary time was r_s_ = 0.3, using a different indicator of prolonged sitting, i.e., total number of breaks [38].

Importantly, the other questions studied also correlated significantly (*p* < 0.001) with total stationary time and prolonged stationary time as assessed by accelerometer, achieving similar Spearman’s rho values (0.4–0.5). This study thus added important information regarding the Katzmarzyk question that was used in a groundbreaking paper associating sedentary time with health, where good predictive validity for 12-year mortality was shown [3]. However, that classical question has not previously been validated for concurrence against device-assessed stationary behavior. Results in the present paper indicated that the question described by Katzmarzyk et al. indeed correlated significantly (*p* < 0.001) with total stationary time (r_s_ = 0.53) and prolonged stationary time (r_s_ = 0.46) assessed by accelerometer.

Earlier validation studies of IPAQ only investigated the correlation with total sedentary time assessed by accelerometer (<100 cpm) [17], while this study also demonstrated a significant correlation (r_s_ = 0.41) with prolonged stationary time. The present study was based on a greater number of participants in another setting (country), consequently strengthening the concurrent validity of the IPAQ sitting item. The correlation with total stationary time assessed by accelerometer was r_s_ = 0.44 in this study, which was somewhat higher than Rosenberger et al. found (r_s_ = 0.34) [17]. Earlier studies have indicated that domain-specific questions with multiple items assess sitting time more accurately than global single-item questionnaires, and therefore might be better for prevalence and surveillance studies [37]. A recent study also suggested that it might be better to ask about time spent in seated activities instead of sitting per se [39]. However, when both types of questions were validated simultaneously in our study, we found that the single-item global questions (sitting) and a domain-specific question (seated activities) with multiple items correlated with accelerometer-assessed stationary behavior to the same degree.

Importantly, all four self-reported questions underestimated time spent sitting, compared to accelerometer-assessed stationary behavior (4–6 h/day versus >8 h). This was in line with other studies, which have often shown an underestimation of 40–50%, or 5–6 h self-reported compared to 9–10 h of device-assessed sedentary time [36]. This is important to remember when SED-GIH is used for screening patients. However, despite the underestimation of sitting hours, the SED-GIH has acceptable sensitivity to identify individuals with a high degree of stationary behavior as assessed by accelerometer.

A strength of the present study was the population sample, consisting of a large number of randomly selected adults (20–67 years), both women (69%) and men, with a wide span of BMI. The study populations in many previous validation studies have been homogenous convenience samples with relatively few participants [19,38]. Accelerometry was used as reference method, and is considered by many to be the standard for assessing both PA and time spent stationary under free-living conditions. The broad statistical approach in this study was also a strength. The chosen tests evaluated not only simple correlations, but also confounding, interaction, linear associations, as well as the discriminative capacity of the SED-GIH question. The inclusion of three other common and validated questions regarding sedentary behaviors was also a strength, as it added to the external validity of the results.

There are, however, some limitations to using accelerometer-assessed stationary time to evaluate the concurrent validity of self-reported sitting. There are differences in the classification of stationary behavior, as the questionnaire included sitting time and the accelerometer cannot distinguish between sitting, lying down, and standing still. Additionally, the question asked for sitting during a whole day, while the accelerometer prerequisite was only ten hours a day. These limitations might explain some of the differences in sedentary time between the two methods, as the misclassification might overestimate accelerometer-assessed time. To adjust for different time periods in the analysis, we standardized the wear time of accelerometer to 16 h/day, resulting in more minutes in both stationary time and prolonged stationary time in the standardized values vs. unstandardized.

Further evaluation of the SED-GIH is needed in form of studies on test–retest reliability and on the responsiveness and sensitiveness to changes in sedentary behavior over time to enable evaluation of interventions. Studies are also needed to validate this question in different populations, as well as in people with different diseases or disabilities.

The results of the present study have several important implications. From a health-promoting perspective, the difference between sitting still and standing still is limited. Rather, larger effects on glucose and insulin levels, for example, are observed between sitting/standing on the one hand and walking on the other [10,11,40]. Thus, the greater public health interest might therefore be to distinguish these stationary from any ambulatory behaviors. Importantly, the present SED-GIH question offers a simple, feasible method to assess stationary behavior in clinical practice. As an alternative to accelerometry, SED-GIH is also useful for identifying individuals with high levels of sedentary behavior. For epidemiological studies, categorical global questions should be used along with, whenever possible, objective assessment of sedentary behaviors and the whole PA pattern, i.e., accelerometry.

## 5. Conclusions

The newly developed single-item question, SED-GIH, is valid for assessment of stationary behavior, showing a moderate concurrent validity, and it is at least as good as existing questionnaires. It may be used in health care for screening an individual’s sitting time or the risk behavior of prolonged sitting time. This has important clinical implications, as simple assessment tools are needed in clinical practice and research on stationary behavior.

## Figures and Tables

**Figure 1 ijerph-16-04766-f001:**
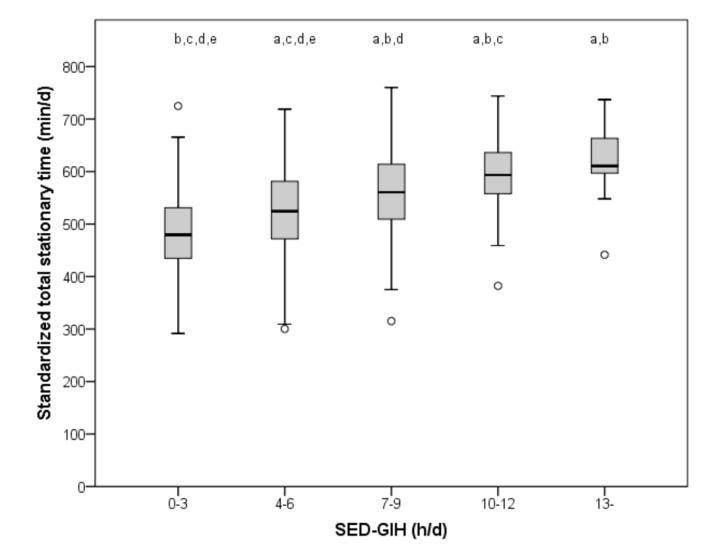
Accelerometer assessed standardized total stationary time across categories of self-reported sitting time (SED-GIH). Significant differences in standardized total sedentary time across the different answer categories (*p* < 0.001): (a) mean difference from 0–3 h, (b) 4–6 h, (c) 7–9 h, (d) 10–12 h, and (e) 13+ h. With SED-GIH answer Alternative 1 as reference (480 min), Alternative 2 corresponded to an increase of in median 44 min, Alternative 3 to 80 min, Alternative 4 to 114 min, and Alternative 5 to 131 min of standardized daily stationary time assessed by accelerometer.

**Figure 2 ijerph-16-04766-f002:**
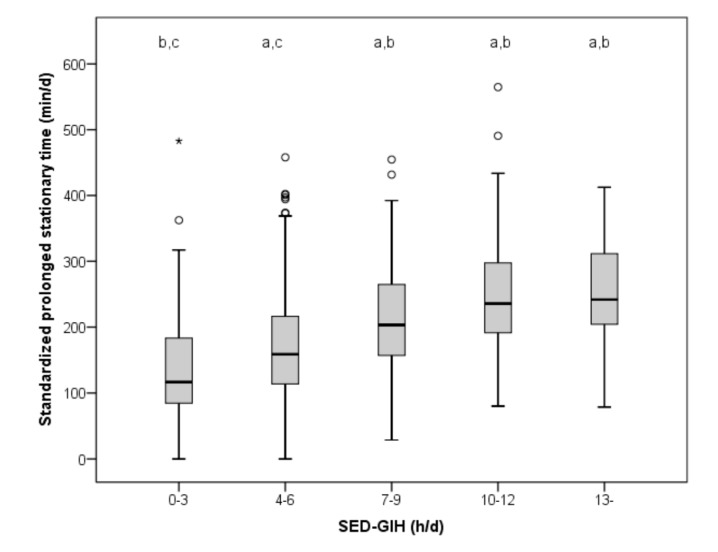
Accelerometer-assessed standardized prolonged stationary time across categories of self-reported sitting time (SED-GIH). Differences in standardized prolonged sedentary time across the different answer categories (*p* < 0.001): (a) mean difference from 0–3 h, (b) from 4–6 h, and (c) from 7–9 h, 10–12 h, and 13+ h. With SED-GIH answer Alternative 1 as reference (117 min), Alternative 2 corresponded to an increase of 42 min in median, Alternative 3 to 87 min, Alternative 4 to 119 min, and Alternative 5 to 125 min of standardized daily stationary time assessed by accelerometer.

**Figure 3 ijerph-16-04766-f003:**
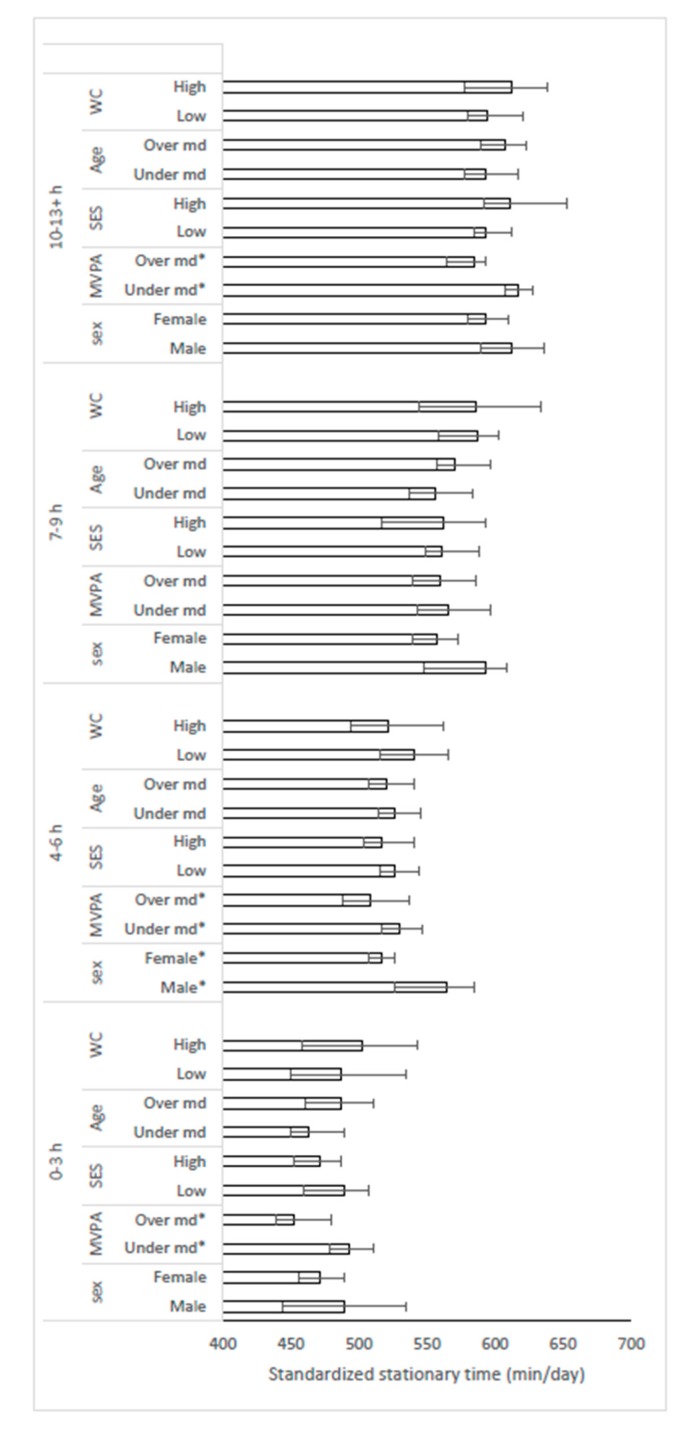
Comparison of mean (95% CI) device-assessed stationary time between selected strata across the response categories. The two upper categories were merged due to low number (*n* = 16) in the 13+ h category. Stationary time was different between MVPA strata in three of four response categories. * = significant difference between strata. MVPA: moderate to vigorous physical activity; SES: education; WC: waist circumference; md: median.

**Table 1 ijerph-16-04766-t001:** Stationary time and physical activity level for participants included in the concurrent validity analysis (top) and the convergent validity analysis (bottom). All data are presented as median (Q1–Q3).

**Concurrent Validity Analysis**	**All (*n* = 711)**	**Women (*n* = 489)**	**Men (*n* = 222)**	***p*-Value ***
SED-GIH ^c^ (*n* = 711)	2 (2–3)	2 (2–3)	3 (2–4)	<0.001
Accelerometer data				
Stationary time (min/day)	492 (433–546)	480 (427–532)	525 (459–578)	<0.001
Standardized ^a^ stationary time (min/day)	540 (481–601)	529 (470–582)	577 (497–625)	<0.001
Prolonged ^b^ stationary time (min/day)	160 (109–224)	147 (99–201)	201 (142–262)	<0.001
Standardized ^a^ prolonged stationary time (min/day)	177 (118–244)	161 (107–219)	217 (158–281)	<0.001
MVPA (min/day)	36 (23–54)	34 (23–51)	41 (25–60)	0.001
**Convergent Validity Analysis**	**All (*n* = 560)**	**Women (*n* = 367)**	**Men (*n* = 193)**	***p*-Value ***
SED-GIH ^c^	2 (2–3)	2 (2–3)	3 (2–4)	<0.001
Katzmarzyk ^c^	2 (2–3)	2 (2–3)	3 (2–3)	<0.001
IPAQ (min)	360 (240–480)	330 (210–480)	360 (270–555)	0.001
Marshall (min)	415 (270–560)	365 (240–539)	480 (330–603)	<0.001
Accelerometer data				<0.001
Stationary time (min/day)	502 (440–553)	486 (433–538)	528 (467–584)	<0.001
Standardized ^a^ stationary time (min/day)	548 (486–606)	533 (471–588)	582 (510–627)	<0.001
Prolonged ^b^ stationary time (min/day)	168 (115–233)	153 (100–208)	201 (145–258)	<0.001
Standardized ^a^ prolonged stationary time (min/day)	184 (125–251)	169 (111–231)	218 (163–279)	<0.001

* Difference between women and men (Mann–Whitney U Test). MVPA: moderate to vigorous physical activity. ^a^ Standardized data to 16 h of wear time (960 min/day) in stationary and prolonged stationary time. ^b^ Prolonged stationary time: total stationary time in bouts ≥20 min; ^c^ SED-GIH and the question used by Katzmarzyk [3] are categorical with seven and five answer alternatives, respectively. IPAQ is an open-ended question [17]. Marshall is the sum of the domain-specific sitting question on weekdays [18,19].

**Table 2 ijerph-16-04766-t002:** Frequency of over-reporting, correct reporting, and under-reporting in the different response categories of the SED-GIH question and in selected strata.

		Over-Reporting	Correct Reporting	Under-Reporting
SED-GIH	0–3 h	-	0	143 (100%)
	4–6 h	0	13 (5%)	266 (95%)
	7–9 h	2 (1%)	88 (52%)	79 (47%)
	10+ h ^a^	37 (31%)	83 (69%)	-
SES	Low	30(7%)	114 (26%)	294 (67%)
	High	39 (3%)	69 (26%)	186 (71%)
Gender	Male *	17 (8%)	71 (32%)	134 (60%)
	Female *	22 (5%)	113 (23%)	354 (72%)
MVPA	Under median *	13 (3%)	97 (24%)	293 (73%)
	Over median *	26 (8%)	87 (28%)	195 (63%)
Age	Under median *	25 (7%)	104 (29%)	235 (65%)
	Over median *	14 (4%)	80 (23%)	253 (73%)
WC	Low	20 (7%)	77 (27%)	191 (66%)
	High	6 (6%)	33 (34%)	57 (59%)

All data is presented as number, *n* (%). * Frequencies were different across strata of gender, MVPA and age, chi^2^
*p* < 0.05. SES: education; MVPA: moderate to vigorous physical activity; WC: waist circumference. ^a^ The two upper categories were merged due to low number (*n* = 16) in the 13+ h category.

**Table 3 ijerph-16-04766-t003:** Correlation between different self-reported sedentary time questions ^a^ and accelerometer-assessed standardized stationary time.

Varieties	Stationary (min/day)	Prolonged Stationary ^b^ (min/day)	SED-GIH	Katzmarzyk	IPAQ (min)	Marshall (min)
SED-GIH	0.48 ^c^	0.44 ^c^	-	0.89 ^e^	0.83 ^e^	0.72 ^d^
Katzmarzyk	0.53 ^d^	0.46 ^c^	0.89 ^e^	-	0.72 ^d^	0.72 ^d^
IPAQ (min)	0.44 ^c^	0.41 ^c^	0.83 ^e^	0.72 ^d^	-	0.70 ^d^
Marshall (min)	0.48 ^c^	0.46 ^c^	0.72 ^d^	0.72 ^d^	0.70 ^d^	-

All results in Spearman’s rho, except for SED-GIH vs. Katzmarzyk where gamma correlation was used. All *p* < 0.001. ^a^ SED-GIH and the question used by Katzmarzyk [3] are categorical, with seven and five answer alternatives, respectively. IPAQ is an open-ended question [17]. Marshall is the sum of the domain-specific sitting question on weekdays [18,19]. ^b^ Prolonged stationary: time in prolonged stationary time (≥20 min). The associations were interpreted as ^c^ moderate (Spearman’s rho 0.3–0.5), ^d^ strong (Spearman’s rho 0.5–0.8), or ^e^ very strong (Spearman’s rho 0.8–1.0).

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
