# Peer review of "The SED-GIH: A Single-Item Question for Assessment of Stationary Behavior—A Study of Concurrent and Convergent Validity"

_ijerph, 2019, doi:10.3390/ijerph16234766_

Round 1

Reviewer 1 Report

General comment.
The study is nicely thought out, executed and written. A minor comment is that the population were Swedish and this is a limitation. What about children?
In addition, it is not clear how they expect to use actigraph as a measure for sedentary behaviour when the references for 22 and 23 must specify its use and validity for measuring sedentary behaviours. This is needed to avoid the blind leading the blind in this research paper!
There needs to be more details about the sampling procedures of the two studies. Are they random? Convenience? What were the context of the surveys?
The authors also state in L55-6, there are no validated measures, and L57 have low to moderate concurrent validity. Yet the authors use two other measures to give validation. Isn't that a bit strange to do that? Please provide some information at to why this may be a useful method given other two instruments are potentially flawed.

Another comment is the use of terminology. There is much debate in using the term objectively. This is problematic as it is not as objective as a set of scales when reporting weight (rather than self reporting). Accelerometers still require a fair amount of validation and processing before it gives the results, based on many factors such as most, counts, averaging, smoothing, epoch, etc. So to use the term objectively is not entirely objective. Objective should mean that if one person measures it they will get the same result no matter how they use it. They currently don't depending on the epoch, smoothing, or whatever. It would be more straight forward to use throughout the paper the terms self report and accelerometer based or device based.
Given the rights based approach to data dissemination, I would suggest the authors look into the data and confound also for disability.

Specific comments
L46-53, is not clear enough. It seems confusing to describe stationary, in this study. What is the purpose of describing stationary, mets, and walking?
L20, write gih in full the first time.
L72, state what these two cross sectional studies were.
Please refer to this recently published paper

Is sitting invisible? Exploring how people mentally represent sitting by Benjamin Gardner, Stuart Flint, Amanda L. Rebar, Stephen Dewitt, Sahana K. Quail, Helen Whall and Lee SmithInternational Journal of Behavioral Nutrition and Physical Activity 2019, 16:85

Table 1, it is not clear what the values represent, there are so many types of values ranges and how they may work together or contradict each other. Put the asterisk on the all column and not that men. The level of significance is too high for this type of sample size. In fact, a better idea would be to report all p values. Also please provide effect sizes.

L281, where can I find this result of Katzmsrzyk in the paper?
L305 gold standard requires referencing, or removing.
L318-20, avoid single sentence paragraph

Reviewer 2 Report

The current study evaluated a new single-item global question of total sitting time 248 can be used as an indirect mean or proxy for total stationary time. From my point of view, this topic is novel and more research like these are needed. Minor comments are shown below.

Introduction:

- …Existing questions show a low to moderate concurrent validity 57 when compared to objective assessment methods for sedentary behaviour. Please, clarify the population and more data related to this studies.

- It is necessary to strengthen the introduction with some more information about why it is necessary to validate SED questions.

Method:

- ...Both SED-GIH and the question developed by Katzmarkzyk the answer alternatives were re-119 arranged in order to go from low sitting time to high. Regarding both the SED-GIH and the question 120 from Katzmarzyk, the answer categories indicating the most sitting time were chosen by none or few 121 participants. For both questions the two highest categories were therefore merged. For the same 122 reason, the two categories from the SED-GIH with lowest sitting time were also merged. Please, could the authors make this paragraph more understandable?.

- ...In this study 126 the standardized wear-time was set to 16 hours (allowing for an 8 hour sleep period per full day). Please, the authors are asked to discuss this point by comparing it with previous studies in the discussion.

Discussion:

- ... In addition, the SED-GIH correlates significantly (p<0.01) with three other commonly used and 253 earlier validated questions (rs between 0.72 - 0.89), showing its convergent validity. Please, clarify specific differences between studies.

- ... This information points to the public health relevance of the SED-GIH, 262 single-item question, as prolonged stationary time, is increasingly important to assess in clinical 263 practice. Clarify why?.

- ... A strength of the present study is the population sample consisting of a large number of 301 randomly selected adults (20-67 years), both women and men, and with a wide span of BMI. It would be interesting to show that there was a high percentage of women vs. men.

Round 2

Reviewer 1 Report

The authors have responded to many of the initial comments appropriately. A few more suggestions are outlined below.

The term sed-gih currently has no international meaning. First sed, is an abbreviation, and not an acronym. Acronym would be stronger. Second, gih is only known in Sweden, and is an abbreviation for the Swedish. No one else really knows this, and Swedish language is not the dominant language in academia. It would be more appropriate for the authors to come up with a better name that could then be translated into an international context.
It is a single item, it is related to stationary behaviours, it is a scale, to me, a simpler form would just be SSS, or the 3S.

Having read the studies from which they have come from, it is very important to highlight this information more accurately. In your discussions, refer that this is used for working age populations and the type of workers. Pull more studies from other workers to see how these results may go beyond the current sample of workers. Try to avoid over generalising the results. This would provide a good paragraph in the discussions, and not just a single sentence in the limitations.

The authors need to acknowledge the other data sources in the acknowledgement and state where the data can be found. Moreover, as recommended by Weston et al, 2019 (
https://doi.org/10.1177/2515245919848684), more information on the secondary analysis is needed. First, provide a link for the main study. Second provide a brief summary of the way the data were collected. There is an issue with regards to the accuracies in data collection without this information. This could compromise the results.

I am surprised that the authors do not report the effect size. With sample sizes like what the authors have used, the significance level is only just one small fraction of understanding the statistical findings. The authors are encouraged to report the effect size in the last table. Moreover, concerning the statistical reporting, only using p>.01 instead of the standard .05 needs explanation and referencing for why this is chosen. Finally, open science publication is designed for replication of on doubt. Therefore, kindly provide all the p values and not just asterisk or bold when findings are below a threshold. The authors are responsible for being as transparent as possible, and should therefore report all p values. For more info, kindly review the following pages (as an example); https://support.jmir.org/hc/en-us/articles/360019690851-Guidelines-for-reporting-statistics
